# Highly Efficient One-Pot Synthesis of Hexakis(*m*-phenyleneimine) Macrocyle Cm6 and the Thermostimulated Self-Healing Property through Dynamic Covalent Chemistry

**DOI:** 10.3390/polym15173542

**Published:** 2023-08-25

**Authors:** Toshihiko Matsumoto

**Affiliations:** Department of Industrial Chemistry, Graduate School of Engineering, Tokyo Polytechnic University, Atsugi 243-0297, Japan; matumoto@chem.t-kougei.ac.jp; Tel.: +81-(0)46-270-2929

**Keywords:** dynamic covalent polymer, Schiff base, azomethine, imine-based macrocycles, water-mediated cyclic–linear conversion, single one-pot procedure, self-healing property, precipitation-driven cyclization, π-stacked columnar aggregates

## Abstract

Highly efficient one-pot synthesis of hexakis(*m*-phenyleneimine) macrocycle Cm6 from acetalprotected AB-type monomer, *m*-aminobenzaldehyde diethylacetal, was successfully achieved based on imine dynamic covalent chemistry and precipitation-driven cyclization. The structure of Cm6 in the solid state was determined using CP/MAS NMR, X-ray single crystallographic analysis, and WAXD. Macrocycle Cm6 is composed of six phenylene and imine bonds facing the same direction, with nitrogen atoms arranged on the outside of the ring, and has a chair conformation, as predicted from DFT calculation. The macrocycle forms π-stacked columnar aggregates and hexagonally closest-packed structure. The cyclization process was investigated using MALDI-TOF MS and NMR. A mechanism of precipitation-driven cyclization based on imine dynamic covalent chemistry and π-stacked columnar aggregation is proposed. Both the nature of imine linkage and the shape anisotropy of the macrocycle played an important role in the single one-pot synthesis. The water-mediated mutual conversion between macrocycle Cm6 and linear oligomers driven by thermal stimulation was analyzed using MALDI-TOF MS and GPC methods. Macrocycle Cm6 with a dynamic covalent imine bond exhibited self-healing properties when stimulated using heat.

## 1. Introduction

The use of shape-persistent macrocycles as functional materials has benefited from recent advancements in their preparative methods [1,2,3,4,5,6,7]. Among macrocyclic compounds, imine macrocycles constitute an important and diverse class of compounds, which have applications in catalysis, recognition, separation, and medical diagnostics [8,9]. Traditionally, the widespread utilization of macromolecules is hindered by tedious preparation, often requiring dilute conditions, small scales, difficult separations, and low overall yields. The efficient preparation of functionalized shape-persistent macrocycles has been a challenging task.

In the past two decades, many efficient synthesis methods for shape-persistent macrocycles through dynamic covalent chemistry (DCC) have been reported. Using a Mo(VI) alkylidyne metathesis catalyst, phenyleneethynylene macrocycles were prepared in high yields from 1,3-dialkynylbenzene derivatives [10,11]. In this reaction, the precipitation of the macrocyclic products from the reaction solvent is the driving force for the reaction. A series of hetero-sequenced shape-persistent imine macrocycles were synthesized through one-pot imine dynamic covalent chemistry resulting in good-to-excellent yields from readily accessible starting materials [12]. The recent implementation of methodologies wherein the bond formation event is under thermodynamic control, otherwise known as dynamic covalent chemistry, has enabled a diverse array of macrocycles to be generated from simple precursors in a less synthetically demanding manner [13,14,15,16,17,18].

In general, imine-based macrocycles are prepared through AA-BB-type polycondensations from dialdehydes and diamines. The [n + n] condensation may result in the formation of various macrocyclic products, such as [2 + 2], [3 + 3], etc., macrocycles as well as linear oligomeric and polymeric imines. The selectivity and the yields are enhanced dramatically in condensation reactions templated by transition metal ions such as zinc(II), copper(II), or nickel(II) [19,20,21,22,23,24,25,26]. J. Lisowski reported that the [3 + 3] macrocycles were sometimes obtained in high yields via direct condensation without a metal template in the condensation of aromatic dialdehydes with chiral diamines such as 1,2-*trans*-diaminocyclohexane [9]. J.-M. Lehn et al. investigated the self-assembly and self-sorting behavior of dynamic covalent organic architectures, which facilitate the parallel generation of multiple discrete products in a single one-pot procedure [27]. They reported the self-assembly of covalent organic macrocycles and macrobicyclic cages from dialdehyde and polyamine components via multiple [2 + 2] and [3 + 2] polyimine condensations. Imine chemistry is among the most well-established reversible reactions and has also been a main synthetic tool. Various shape-persistent macrocycles and covalent organic polyhedrons have been efficiently constructed in one step through dynamic imine chemistry. W. Zhang and co-workers prepared imine-linked porous polymer networks, which exhibit permanent porosity with high specific surface areas. Their most recent contribution is the discovery of a recyclable polyimine material whose self-healing can be activated simply by heating or water treatment [16]. D. Zhang and co-workers developed covalent polymeric networks composed of imine cross-linkages, which exhibit malleability and self-healing characteristics [28].

Hughes and co-workers reported on the efficient synthesis of an imine-based macrocycle from an AB-type monomer [29]. They synthesized a novel *ortho*-phenylene-*para*-phenyleneimine macrocycle through one-pot reduction and the cyclooligomerization of 2-(4-nitrophenyl)benzaldehyde. Specifically, Fe(0) and aqueous HCl reduce nitroaldehyde to the AB-type monomer aminoaldehyde, which then undergoes spontaneous macrocyclization, leading to moderate yields of the macrocycle. Recently, K. Mori et al. developed a concise access to C3-symmetric imine-linked macrocycles from AB-type monomers with aldehyde group and Boc-protected amine [30]. When the monomers were treated with an excess amount of concentrated HCl in 1,4-dioxane, the detachment of the Boc group followed by a trimerization reaction via imine formation proceeded smoothly to afford C3-symmmetric imine-linked macrocycles in good chemical yields.

Our research focuses on the design and synthesis of discrete molecular architectures such as macrocycles through a simple one-pot procedure and their self-healing property tuned using thermal stimulation.

## 2. Materials and Methods

### 2.1. Materials and Instruments

*m*-Nitrobenzaldehyde, 5% palladium on activated carbon (5%-Pd/C), sulfuric acid, tetrahydrofuran, and ethanol were obtained from FUJIFILM Wako Pure Chemical Corporation (Tokyo, Japan). Triethoxymethane was purchased from Tokyo Chemical Industry Co., Ltd. (Tokyo, Japan). All the reagents were used without further purification.

The ^1^H and ^13^C NMR spectra were obtained using a JEOL JNM-LA500 spectrophotometer (Tokyo, Japan). The proton signals in the ^1^H-NMR spectrum were assigned from the H,H- and C,H-COSY spectra. The measurement conditions for CP/MAS ^13^C-NMR were a resonance frequency of 15 MHz, a 90° pulse of 5.5 μs, a ^1^H contact time of 2–3 ms, a pulse repetition time of 8–12 s, and a sample tube rotation rate of about 2.1 kHz. Infrared spectra were recorded using a JASCO VALOR III Fourier transform spectrometer (Tokyo, Japan). The single-crystal X-ray structure analysis of Cm6 recrystallized from water-containing tetrahydrofuran was carried out on a Bruker SMART APEX-II ULTRA CCD diffractometer (Bruker Japan Co., Yokohama, Japan) with Mo Kα radiation and a 12 kW rotating anode generator at 100(1) K. The calculations were performed using the program system SHELXS-97. Full-matrix least-squares refinement on F^2^ with all non-hydrogen atoms anisotropic was carried out. MALDI-TOF MS measurements were conducted with an Applied Biosystems Voyager-DETEPRO-T spectrometer (Waltham, MA, USA). Gel permeation chromatography (GPC) measurements were performed at 40 °C on a JASCO ChromNAV system (Tokyo, Japan) consisting of an isocratic JASCO PU-2080 Plus pump, a set of three columns (Shodex KF-801 + KF-802 × 2), and a JASCO Intelligent UV Detector UV-970 (Tokyo, Japan) and a JASCO Intelligent RI Detector RI-2031 Plus (Tokyo, Japan). TEM (JEM 2000EXII, JEOL Co., Tokyo, Japan) with 20 kV acceleration voltage was used to observe the morphology and obtain an electron diffraction (ED) image of the macrocycle Cm6. The sample for the TEM observation was prepared by placing water suspensions of the fine crystals on a Cu grid and then air-drying.

### 2.2. Preparation of Monomer: m-Aminobenzaldehyde Diethylacetal

#### 2.2.1. Synthesis of *m*-Nitrobenzaldehyde Diethylacetal (**1**)

In a 500 mL pear-shaped flask with an Allihn condenser, *m*-nitrobenzaldehyde (50.0 g, 330.8 mmol), triethoxymethane (49.0 g, 330.8 mmol), EtOH (solvent, 250 mL), and H_2_SO_4_ (200 μL) were added and refluxed for 27 h. After refluxing, the reaction mixture was cooled to room temperature in an ice bath, neutralized with NaHCO_3_ powder, and filtered off. The solvent was removed via evaporation to yield a yellow liquid. The crude product was purified using distillation under reduced pressure to yield *m*-nitrobenzaldehyde diethylacetal (**1**) as a pale-yellow liquid (108.5 °C/1.0 mmHg). Yield: 65.1 g, 87.3%.

IR (liquid film, cm^−1^); 2987~2883 (acetal C-H), 1531 (NO_2_), 1350 (NO_2_), 1200 and 1115 (acetal C-O-C). ^1^H-NMR (CDCl_3_, δ); 1.27 (*t*, *J*_9,8_ = 7.0 Hz, 6H, H-9), 3.56–3.67 (*m*, 4H, H-8), 5.60 (*s*, 1H, H-7), 7.56 (*dd*, *J*_5,4_ = *J*_5,6_ = 7.8 Hz, 1H, H-5), 7.84 (*d*, *J*_6,5_ = 7.8 Hz, 1H, H-6), 8.17 (*d*, *J*_4,5_ = 7.8 Hz, 1H, H-4), 8.35 (*s*, 1H, H-2). ^13^C-NMR (CDCl_3_, δ); 15.20 (C-9), 61.42 (C-8), 100.16 (C-7), 121.92 (C-4), 123.26 (C-2), 129.28 (C-5), 132.98 (C-6), 141.66 (C-1), 148.38 (C-3) (Appendix A).

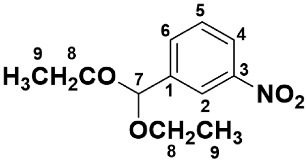



#### 2.2.2. Synthesis of *m*-Aminobenzaldehyde Diethylacetal (**2**) via Catalytic Hydrogenation of *m*-Nitrobenzaldehyde Diethylacetal

The nitro-compound (**1**) (10.0 g, 46.6 mmol), 5%-Pd/C (0.550 g, 0.258 mmol as Pd atom), and tetrahydrofuran (solvent, 60 mL) were placed in a 100 mL glass autoclave (TEM-V100, TAIATSU TECHNO Corp., Tokyo, Japan) equipped with a mechanical stirrer and a thermocouple. Hydrogen gas was supplied to the apparatus for 3.5 h while maintaining a constant pressure of 0.5 MPa. The catalyst was filtered off through Celite^®^, and the filtrate was dried with MgSO_4_ and then condensed to dryness by evaporation. Yield: 8.78 g, 100.8%.

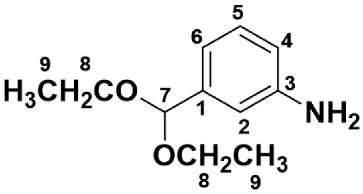



An aliquot of the crude product (3.0 g) was purified using a Kugelrohr distillation apparatus (Sibata Glass Tube Oven GTO-350RD, Tokyo, Japan) to yield *m*-aminobenzaldehyde diethylacetal (**2**) as a pale-yellow liquid (130 °C/1.0 mmHg). Yield: 2.17 g, 72.3%. 

IR (liquid film, cm^−1^); 3456 (NH_2_ *as*), 3365 (NH_2_, *sy*), 2978~2883 (acetal C-H), 1623 (aromatic amine), 1200 and 1113 (acetal C-O-C *as* and *sy*). ^1^H-NMR (CDCl_3_, δ); 1.20 (*t*, *J*_9,8_ = 7.3 Hz, 6H, H-9), 3.48–3.64 (*m*, 4H, H-8), 3.70 (*bs*, 2H, NH_2_), 5.38 (*s*, 1H, H-7), 6.57 (*d*, *J*_4,5_ = 7.9 Hz, 1H, H-4), 6.78 (*s*, 1H, H-7), 6.82 (*d*, *J*_6,5_ = 7.3 Hz, 1H, H-6), 7.10 (*dd*, *J*_5,4_ = *J*_5,6_ = 7.9 Hz, 1H, H-5). ^13^C-NMR (CDCl_3_, δ); 15.22 (C-9), 61.01 (C-8), 101.74 (C-7), 113.23 (C-2), 115.00 (C-4), 116.64 (C-6), 129.00 (C-5), 140.26 (C-1), 146.77 (C-3) (Appendix A).

### 2.3. Polymerization: Synthesis of Hexakis(m-phenyleneimine) Macrocycle (Cm6)

*m*-Aminobenzaldehyde diethylacetal (**2**), tetrahydrofuran, and water (initiator) were placed into a glass bottle, and the mixture was magnetically stirred at room temperature until white precipitates appeared. The precipitates were filtered off and dried at 80 °C for 8 h under reduced pressure. A typical polymerization condition is as follows: *m*-aminobenzaldehyde (**2**) (1.0 g, 5.1 mmol); solvent (4.0 mL, THF/H_2_O = 7/1 (*v*/*v*)); concentration (1.28 M); room temperature; 6 days. The resulting solid (isolated yield: 82%) was recrystallized from tetrahydrofuran using a slow solvent-evaporation technique to yield a white crystalline material. IR (KBr, cm^−1^): 1624 (νC=N), 1597 (νC=C), 1574, 1237, 1170, 791, 687, 417. UV–Vis (THF, nm): 330 (sh), 274, 222. Elemental Anal. Calcd. for C_42_H_30_N_6_·0.5C_4_H_8_O (THF)·3H_2_O: C, 74.56; H, 5.69; N, 11.86. Found: C, 75.8; H, 5.66; N, 12.0 (Appendix A).

### 2.4. Self-Healing Analysis of Cm6 Using MALDI-TOF MS and GPC

In a 200 mL three-necked flask with an Allihn condenser, *m*-aminobenzaldehyde diethylacetal (10.0 g, 51.3 mmol) and a mixture of tetrahydrofuran (80 mL) and water (20 mL) were added. The solution was refluxed for 8h with magnetic stirring, then the reaction mixture was cooled to room temperature. An aliquot (10 µL) was collected at a certain interval and diluted with tetrahydrofuran (2 mL), and the solution was filtered through a PTFE membrane filter (pore size 0.45 µm) for MALDI-TOF MS and GPC measurements.

MALDI-TOF MS: α-Cyano-4-hydroxy cinnamic acid was used as the matrix and as the external reference for spectrum calibration. The sample with or without the matrix was placed on the MALDI-TOF MS target (3 mm diameter) steel plate. After the evaporation of the solvent, the MALDI-TOF MS target was introduced into the spectrometer. The irradiation source was a pulsed nitrogen laser with a wavelength of 337 nm. The length of one laser’s pulse was 3 ns. Measurements were carried out using the following conditions: polarity positive; flight path reflection; 20 kV acceleration voltage; 50 pulses per spectrum; and acquisition mass range 100–5000 Da. The delayed extraction technique was used applying a delay time of 85 ns. The software MALDI-TOF MS Ver. 1 was used for data treatment. The spectra precision was of ±1 Da.

GPC: A sample solution (10 μL) was injected into the GPC instrument, and tetrahydrofuran was used as the eluent at a flow rate of 0.6 mL/min at 40 °C. Polystyrene standards were utilized for the calibration of molecular weights.

## 3. Results and Discussion

### 3.1. Monomer Synthesis

An AB-type monomer having both amino and aldehyde groups like *m*-aminobenzaldehyde (**3** in Figure 1) undergoes spontaneous self-polycondensation at room temperature to yield the imine oligomers or polyimines, which makes it difficult to control the polymerization. Herein, in order to avoid the undesirable spontaneous polymerization, the aldehyde group was protected with diethylacetal having low reactivity toward the amino group. The synthetic route of *m*-aminobenzaldehyde diethylacetal (**2**) is illustrated in Figure 1. The starting material, *m*-nitrobenzaldehyde, was reacted with triethoxymethane in ethanol in the presence of catalytic amounts of H_2_SO_4_ to yield *m*-nitobenzaldehyde diethylacetal (**1**). The Pd-catalyzed hydrogenation of the nitro-compound (**1**) gave *m*-aminobenzaldehyde diethylacetal (**2**). The hydrogenation is an exothermic reaction, so the temperature of the reaction solution increased quickly and reached 70 °C after 0.5 h. Afterward, the temperature gradually decreased to room temperature (ca. 3 h). The product was purified using a Kugelrohr distillation apparatus, and the chemical structure was identified with IR and NMR spectroscopies.

### 3.2. Polymerization: Synthesis of Macrocycle Cm6

*m*-Aminobenzaldehyde (**2**) (1.0 g, 5.1 mmol) was polymerized in water-containing tetrahydrofuran (4.0 mL, THF/H_2_O = 7/1 (*v*/*v*)) at room temperature under magnetic stirring. Water plays a role in deprotecting the acetal group slowly and regenerating the aldehyde group. The polycondensation between the regenerated aldehyde and an amino group of another monomer immediately occurs; that is, water acts as an initiator of the polymerization. At an earlier stage, the polymerization proceeded homogeneously; however, after approximately one week, white precipitates suddenly appeared. This process was analyzed via MALDI-TOF MS spectroscopy. The spectra of the reaction mixture are illustrated in Figure 1.

In the spectrum of the solution one day after the reaction started, three major peak series appear. The correct repeat unit mass difference of 103.12 amu is observed between the peaks, which means that the products consist of *m*-phenylene imine oligomers terminated with different end groups. The peak A series are due to linear oligomers terminated with an acetal group and an amino group, and the B series are those terminated with an aldehyde group and an amino group. The C series are assigned to linear oligomers terminated with an amino group and an unknown group or cyclic oligomers with one hemi-aminal linkage; however, it has not been exactly identified yet. Five days later, the intensities of the B and C series increased, whereas those of A series decreased. Six days later, white powdery precipitates suddenly appeared, and in the mass spectrum, a very intense peak was observed at 618.88 amu, which corresponds to hexakis(*m*-phenylene) macrocycle (Cm6), and the isolated yield was more than 70%. The weak peaks of the B and C series and the peak corresponding to the double-ring catenane are also observed. The time until precipitates appeared and the yields under various polymerization conditions are summarized in Table 1. In a range of monomer concentration from 0.05 to 1.28 M, the time until Cm6 precipitates appeared became shorter as the monomer concentration increased. The yields tended to increase with an increase in monomer concentration. This result is of interest as it goes against the common knowledge that macrocycle preparation requires dilute conditions. In polymerizations at concentrations below 0.67 M, it may take more than one month until Cm6 precipitates appear. However, when the polymerization solution (0.51 M) was refluxed for 8 h at first, the Cm6 precipitates appeared after three days. Polymerization solvents other than THF such as dioxane and ethanol were effective for Cm6 synthesis, and one molar acetic acid influenced the synthesis as an initiator. The yields and the time until precipitates appeared were markedly influenced by the ratio of THF and water (Appendix A), which means that a higher polarity of the mixed solvent strengthened the π-stacking aggregation of Cm6.

### 3.3. Structure of Cm6

#### 3.3.1. NMR Analysis

The precipitate was hardly soluble in common organic solvents but very slightly soluble in NMP or THF. The ^1^H-NMR spectrum measured in *N*-methylpyrrolidone (NMP)-*d*_9_ is shown in Figure 2. The imine proton signal is observed around 8.5 ppm. Benzene-ring signals appear in a range from 7.1 to 7.7 ppm. These positions are quite similar to those of benzylideneaniline (model compound), and the upper field shift of the imine and H-2 protons was not observed due to having a ring current effect like porphyrin [31,32]. This result indicates that the macrocycle is not fully conjugated and does not have a wide-area resonance.

Macrocycle Cm6 did not show sufficient solubility to allow solution ^13^C-NMR measurement in available deuterated solvents. The solid-state cross-polarization magic-angle (CP/MAS) ^13^C-NMR spectrum of Cm6 was measured, and the spectrum is illustrated in Figure 3 together with the chemical shift values. As evident in the spectrum, seven signals appear for each carbon of the *m*-phenyleneimine unit structure of Cm6 and are assigned as shown in Figure 3 considering the nuclear Overhauser effect [33]. Despite the sample being sufficiently dried under the vacuum, the signals attributed to the polymerization solvent tetrahydrofuran (THF) appear at 24.4 ppm and 66.9 ppm in the spectrum. As will be mentioned later, single-crystal X-ray analysis revealed that one molecule of THF intercalates between the molecules of the macrocycle Cm6. The FT-IR spectrum also supports the idea that the precipitate is a macrocycle, due to the lack of peaks assigned to terminal groups such as acetal, aldehyde, and amino groups.

#### 3.3.2. Prediction of Optimal Structure Using MO Calculation

The optimal structure of the Cm6 molecule was predicted using ab initio density-functional theory (DFT). There are two possible chemical structures for the macrocycle Cm6, as illustrated in Figure 4. Structures (a) and (b) have nitrogen atoms aligned on the outer and inner rims of the ring, respectively. The rotational energy barrier along a bond axis was calculated using Gaussian 03 (B3LYP, 6-31G(d,p)) software [34]. The energy barrier is 8 kJ/mol for the imine nitrogen-phenylene carbon bond and 41 kcal/mol for the imine carbon-phenylene carbon bond. Thermal interconversion between outer- and inner-rim nitrogen-type isomers may hardly occur at room temperature. An optimized conformation for macrocycle Cm6 was calculated using Gaussian 03 (B3LYP, 6-31G(d,p)) software. The Cm6 molecule has two minimum energy conformations, namely (a) chair and (b) boat forms, similar to cyclohexane (Figure 5a,b). The chair conformation is 7.1 kJ/mol more stable than the boat conformation, and both conformations are interconvertible at room temperature when existing as a single molecule in a vacuum. The structure of two molecules was also optimized (Figure 5c). It is quite interesting that two chair-formed Cm6 molecules aggregate in a slipped–parallel manner through aromatic π-stacked interactions. This indicates that Cm6 molecules stack up and form a columnar assembly.

#### 3.3.3. X-ray Analysis of Cm6 Crystal

The X-ray crystal data for Cm6, acquired from a colorless crystal, are listed in Table 2, along with the structure refinement. The other data including atomic coordinates (Appendix A), bond lengths and angles (Appendix A), anisotropic displacement parameters (Appendix A), and hydrogen coordinates (Appendix A) are described in Appendix A. The ORTEP drawings of Cm6 are shown in Figure 6a,b.

Single-crystal X-ray analysis revealed that the Cm6 molecule adopts an outer-rim configuration with respect to its six nitrogen atoms and has a chair conformation. The Cm6 macrocycles form π-stacked aggregates in the (100) direction of the crystal (Appendix A) with a tubular channel, as predicted from the DFT calculation of the two-molecule structure. It is interesting to note that one molecule of THF and a cluster of six water molecules are randomly intercalated between macrocycles in a disordered state (Figure 6c,d). In the ORTEP drawing, the averaged structure of the intercalated substance is displayed. The empirical formula and molecular weight corresponding to the unit cell are C_42_H_30_N_6_ + 0.5THF (C_4_H_8_O) + 3H_2_O and 708.84, respectively. 

From the WAXD pattern of finely ground powdered Cm6 (Figure 7), it can be estimated that the macrocycles are π-stacked with a distance of 5.4 Å and form a columnar packing with a cylindrical channel. The outer and inner diameters are estimated to be 16.8 Å and 7 Å, respectively. In the solid state, the columns aggregate with a hexagonally closest-packed structure. WAXD data (2θ and d-value) are listed in Table 3.

#### 3.3.4. TEM-ED Analysis of Cm6

The transmission electron microscopic (TEM) image of Cm6 is displayed in Figure 8a. The area enclosed by the red circle is where electron diffraction (ED) was measured. As described above, it can be inferred from single-crystal X-ray structure analysis that Cm6 has imine nitrogen atoms on the outer rim of the ring, with C-H facing the inside of the ring, forming a pseudo-plane with six benzene rings and adopting a chair-like structure. It is also clear from powder WAXD that these form nanocolumns by stacking in a slipped parallel manner at intervals of 4–5 Å, and that the columns are arranged hexagonally. In this TEM image, needle-like (whisker-like) crystals with a thickness of about 200 nm and a length of several μm are observed. Figure 8b shows the electron diffraction pattern of the part enclosed by a red circle in this needle. A diffraction spot is observed at the position corresponding to 0.38 nm, which is the distance between Cm6 molecules stacked in the crystal length direction. This is consistent with the results of X-ray diffraction.

### 3.4. Possible Mechanism for Quantitative Formation of Macrocycles

The polymerization process analyzed using ^1^H-NMR is illustrated in Figure 2 and Figure 9. Polymerization was carried out in a melt-sealed NMR tube containing *m*-aminobenzaldehyde diethylacetal, deuterated water (D_2_O), deuterated tetrahydrofuran (C_4_D_8_O), and a small amount of light water (H_2_O). Before heating the tube, acetal and phenylene proton signals of the monomer are clearly observed. After heating the tube at 70 °C for 10 min, the signal intensities of the acetal group decrease, and the signals corresponding to aldehyde, imine, and ethanol protons appear. The acetal proton signals diminish, and ethanol signals markedly increase after heating for 40 min. As shown in the spectrum, after heating for 100 min, acetal proton signals practically disappear, and aldehyde and imine proton signals still remain. These facts indicate that, in the first stage, the acetal group is deprotected with water to yield free aldehyde, and then the generated aldehyde condenses with the amino group of another monomer to form the imine linkage. When the NMR tube was allowed to stand for three weeks at room temperature, the Cm6 macrocycle was deposited at the bottom of the tube, as can be seen in Figure 9.

A possible mechanism for the precipitation-driven quantitative formation of macrocycle Cm6 based on MALDI-TOF and NMR analysis is illustrated in Figure 10. The acetal-protected monomer, *m*-aminobenzaldehyde diethylacetal, is slowly deprotected with water acting as a weak acid, which supplies a condensable aldehyde-amino monomer in the polymerization system at a low concentration. The monomers undergo polycondensation to yield linear imine oligomers in an equilibrium state, as shown in Figure 2. A random-coil oligomer forms a helical foldamer with a strong intramolecular association of hexameric macrocycles having the same backbone structure. A *trans*-imination (imine metathesis) occurs between two imine linkages or between an amino group and a neighboring imine linkage to yield a macrocycle [28]. 

Shape-anisotropic, nearly planar, and persistent macrocycles are known to aggregate into columnar assemblies in polar solvents driven by aromatic π-stacking. The macrocycle is stabilized due to the free energy gained from the intermolecular and noncovalent interactions upon aggregation, and therefore the aggregates become thermodynamically the most stable species and precipitate [35]. The *trans*-imination equilibrium is shifted in favor of cyclization, and the macrocycle is produced as the predominant product (Appendix A). The reversible (dynamic covalent) nature of the imine linkage plays an important role in the quantitative formation. When the aggregates are heated in the solvent containing water, the macrocycle is converted to a random coil, which is entropically stable [36]. This is an example of dynamic covalent chemistry, which concerns chemical reactions that are carried out reversibly under equilibration conditions. Both the dynamic covalent nature of the imine linkage and the shape anisotropy of the macrocycle play important roles in quantitative formation.

### 3.5. Self-Healing Property 

#### 3.5.1. MALDI-TOF MS Analysis

The mutual conversion between macrocycle Cm6 and linear oligomers driven by thermal stimulation, that is, the thermostimulated self-healing of Cm6, was investigated using MALDI-TOF MS spectroscopy (Figure 11). First, the monomer was polymerized in the mixed solvent of THF and water at reflux temperature for 8 h under magnetic stirring, and then the solution was cooled to room temperature. As can be seen in the mass spectrum, just after cooling, three major peak series A, B, and C corresponding to the imine oligomers, as previously mentioned, appear. After 3 days, the Cm6 macrocycle precipitated, and the solution became turbid. An intense and single peak is observed at 618.48 amu corresponding to Cm6. When the turbid suspension was heated for 30 min at reflux temperature, the solution became clear again. In the mass spectrum, there are two major peak series B and C assigned to the linear imine oligomers, which exhibit a product distribution similar to that obtained by heating the monomer for 8 h. It is interesting that no peaks of A series oligomers with the acetal end groups are observed. The imine linkage of the macrocycle was hydrolyzed to provide linear oligomers, and the linear oligomers reacted with each other with repeating formation–cleavage of the imine linkage. When the clear solution was allowed to stand for 3 days at room temperature under magnetic stirring, it became turbid again, and Cm6 macrocycles appeared as precipitates. This mutual conversion between the macrocycle and the linear oligomers is repeated many times, and it has perfect reversibility. It was found that macrocycle Cm6 has a self-healing property in an aqueous THF solution by thermal stimulation.

#### 3.5.2. GPC Analysis

The thermostimulated self-healing property of macrocycle Cm6 was analyzed using not only MALDI-TOF MS spectroscopy but also gel permeation chromatography (GPC). Macrocycle Cm6 is hardly soluble in THF but can be measured with an ultraviolet detector usually with high sensitivity.

The GPC profiles exhibiting the self-healing process are illustrated in Figure 12. The peak corresponding to Cm6 appears at 37.5 min of retention time. In the GPC chart of the reaction mixture obtained by heating the monomer(*m*-aminobenzaldehyde diethylacetal) at the reflux temperature for 8 h, many peaks corresponding to the linear oligomers, including the monomer, appear. These high-molecular-weight oligomers are also observed as a broad peak in the retention time range of 27–35 min. In the chart of the solution produced by being left for 20 h at room temperature, as the peak intensity of high-molecular-weight linear oligomers decreases, a peak corresponding to the molecule Cm6 appears. As more time passes, the peak intensity of low-molecular-weight oligomers slightly decreases, while that of Cm6 increases. Two weeks later, Cm6 becomes dominant, and its chart becomes similar to (a). At room temperature, linear oligomers with various molecular weights undergo repeated formation and cleavage of imine bonds to converge to thermodynamically stable macrocycle Cm6. When the macrocycle is heated in aqueous THF, it is converted to linear imine oligomers, and this process is reversible. It can be seen that macrocycle Cm6 has the property of self-healing upon thermal stimulation.

## 4. Conclusions

An efficient one-pot synthesis of hexakis(*m*-phenyleneimine) macrocycle Cm6 was successfully achieved through a precipitation-driven cyclization reaction from AB-type monomer *m*-aminobenzaldehyde diethylacetal. The yields tended to increase with an increase in the monomer concentration. The mechanism of quantitative formation was estimated based on MALDI-TOF MS analysis and NMR spectroscopy. The chemical and crystal structures of Cm6 were determined using CP/MAS NMR spectroscopy, X-ray diffraction, and electron diffraction methods. The single-crystal X-ray analysis revealed that the Cm6 molecule adopts an outer rim regarding its six nitrogen atoms and has a chair conformation. Macrocycle Cm6 formed π-stacked aggregates with a tubular channel, as predicted from DFT calculation for the two-molecular structure, which is consistent with the results of electron diffraction. The water-mediated mutual conversion between macrocycle Cm6 and linear oligomers driven by thermal stimulation was analyzed using MALDI-TOF MS and GPC methods. It was found that macrocycle Cm6, which has an imine bond that can be reversibly formed and cleaved in the presence of water, is self-healing when thermally stimulated in the aqueous THF.

## Data Availability

The data presented in this study are available upon request from the author (T.M.).

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
