# Peer review of "Highly Efficient One-Pot Synthesis of Hexakis(m-phenyleneimine) Macrocyle Cm6 and the Thermostimulated Self-Healing Property through Dynamic Covalent Chemistry"

_polymers, 2023, doi:10.3390/polym15173542_

Round 1

Reviewer 1 Report

The authors have presented an interesting paper on the synthesis of hexakis(m-phenyleneimine) macrocycle Cm6  through a precipitation-driven cyclization reaction from AB-type monomer m-aminobenzaldehyde diethylacetalIntroduction. The quality of this paper is quite high. 

There are few improvements to do, such as on the English: "in this rection" 

There is no info on the methods for TEM imaging. More information should be given on how the material was prepared, put on grid, which set up was used etc.

For the GPC method, which machine did you use?

The English is overall very good.

Author Response

You can see Answer Sheet Reviewer 1 as an attached file.

Reviewer 2 Report

In the submitted manuscript, Toshihiko Matsumoto reports the beautiful example of dynamic covalent chemistry allowing the selective formation of the Cm6 macrocycle starting from the self-complementary synthon - m-aminobenzaldehyde, formed by in situ hydrolysis of its acetal derivative. The mechanism of Cm6 formation is studied in detail by NMR and mass spectrometry, and the stabilization obtained due to intermolecular stacking interactions is considered as the driving force of the inherent sorting processes leading to the precipitation of the crystalline phase containing the pure macrocycle. The self-healing behavior of the described dynamic system is also demonstrated.

I believe that the methodological elegance and accuracy of the experiments performed clearly justify the possibility of accepting this work for publication, with only a minor revision of the text required.

Line 73 – “Feo and aqueous HCl” – guess, it should be “Fe(0) and aqueous HCl”

Line ­­­89 – “All the reagents were used without further polymerization” – guess, it should be “All the reagents were used without further purification”

Section 2.2.2 – if possible, indicate please the temperature and pressure used for Kugelrohr distillation

Lines 189-190 “to afford 189 m-nitobenzaldehyde diethylacetal (1) quantitatively” – I think the word “quantitatively” can be removed and the actual yields indicated in the experimental section can be indicated for clarity.

Figure 1 and its description – it is not totally clear if the spectrum (c) corresponds to the supernatant, the precipitate or the entire sample of the inhomogeneous reaction mixture.

Line 232 – “In a range of monomer concentration from 1.28 to 0.05 M, the time until Cm6 precipitates appear becomes shorter as the monomer concentration increases» - I think it is better to rephrase “In a range of monomer concentration from 0.05 to 1.28 M” as this order corresponds to the increase of the concentration.

Section 3.3.3 – I don’t fully understand why the authors discuss possible packings if they have single-crystal X-ray data which clearly gives the packing patterns. I think, the word “possible” can be safely removed.

All issues where the text can be improved are indicaterd in the reviewer report

Author Response

You can see Answer Sheet Reviewer 2 as an attached file.
